# Kinematics and Singularity Analysis of a 7-DOF Redundant Manipulator

**DOI:** 10.3390/s21217257

**Published:** 2021-10-31

**Authors:** Xiaohua Shi, Yu Guo, Xuechan Chen, Ziming Chen, Zhiwei Yang

**Affiliations:** 1School of Mechanical Engineering, Yanshan University, Qinhuangdao 066004, China; xhshi@ysu.edu.cn (X.S.); chenxuechan@foxmail.com (X.C.); chenzm@ysu.edu.cn (Z.C.); yzwrobot@163.com (Z.Y.); 2Parallel Robot and Mechatronic System Laboratory of Hebei Province, Yanshan University, Qinhuangdao 066004, China

**Keywords:** 7-DOF manipulator, inverse kinematics, Jacobian matrix, singularity analysis, singularity avoidance

## Abstract

A new method of kinematic analysis and singularity analysis is proposed for a 7-DOF redundant manipulator with three consecutive parallel axes. First, the redundancy angle is described according to the self-motion characteristics of the manipulator, the position and orientation of the end-effector are separated, and the inverse kinematics of this manipulator is analyzed by geometric methods with the redundancy angle as a constraint. Then, the Jacobian matrix is established to derive the conditions for the kinematic singularities of the robotic arm by using the primitive matrix method and the block matrix method. Then, the kinematic singularities conditions in the joint space are mapped to the Cartesian space, and the singular configuration is described using the end poses and redundancy angles of the robotic arm, and a singularity avoidance method based on the redundancy angles and end pose is proposed. Finally, the correctness and feasibility of the inverse kinematics algorithm and the singularity avoidance method are verified by simulation examples.

## 1. Introduction

With the rapid development of robotics and artificial intelligence technology, robots are increasingly used in many industrial applications, such as automobile manufacturing, mechanical processing, electrical and electronic industries, rubber and plastic industries, etc. Compared with the 6-DOF manipulator, a redundant manipulator with unique self-motion characteristics can accomplish complicated tasks easily, such as obstacle avoidance, singularity avoidance, and torque optimization [1]. The stiffness of the redundant manipulator decreases with the increase of the number of axes, so the 7-DOF manipulator, which is considered the simplest redundant manipulator, attracts the attention of scholars [2,3].

For redundant manipulators’ kinematic problems, a challenging problem is how to select an appropriate solution because the number of inverse kinematics solutions is infinite. Usually, the following methods are used: geometric method [4]; numerical method [5,6,7], such as pseudo-inverse of Jacobian matrix [8], augmented Jacobian matrix [9,10], gradient projection method [11], weighted least-norm solution [12], etc. The kinematic control of redundant manipulators can be achieved by position-based or velocity-based methods. Additionally, the latter is considered the standard approach to derive the inverse kinematics expression of the redundant manipulator [13]. However, compared with the position-based solution, the velocity-based solution exhibits several disadvantages, such as returning only one solution [14,15], requiring the pre-assignment of Cartesian trajectory, accumulation error, etc. [16] Since the number of inverse kinematics solutions of the redundant robot arm is infinite, the position-based closed-form solution is difficult to derive. Usually, the aided parameter, such as redundant angle, is introduced to derive the expression of inverse kinematics [17]. However, how to choose the appropriate joint angle needs to be researched further. A method about the comparison of the workspace proposed by Zaplana et al. is used to solve the problem about the selection of the redundant angle [18]. An elbow angle method with a more obvious geometric meaning is proposed by Kreutz-Delgado et al. [19], which can select the value of the redundant parameters according to the task. A position-based control method for the KUKA LBR (A commercial 7-DOF manipulator from KUKA Corporation) manipulator is proposed by Faria [16], and its configuration is an SRS (spherical joint, revolute joint, spherical joint) humanoid manipulator. Two auxiliary parameters are introduced to deal with the self-motion manifolds, the inverse kinematics is obtained by the geometric method, and the singularity avoidance is analyzed.

The singularity analysis and avoidance are very important for the path planning and movement control of redundant manipulators [20,21]. The Jacobian matrix for the redundant manipulator is a non-square matrix; the singularity condition cannot be obtained by simply finding the determinant of the matrix, so it is difficult to solve it. The Gram–Schmidt decomposition of the Jacobian matrix was used by Podhorodeski et al. [22] to solve the singular configurations of the redundant manipulator with a spherical wrist. Nokleby et al. [23] proposed a method based on screw reciprocity, which can be regarded as a general method for the singularity analysis of redundant manipulators. Kong et al. [24] proposed an approach of dependent-screw suppression to analyze the singularity of a 7-DOF redundant manipulator Canadarm2. Xu et al. [25,26] discussed the singularity condition of Canadarm2 by using the elementary matrix of Jacobian. To solve the singularity avoidance problem, the general methods are damped least variance (DLS) [27], singular value decomposition (SVD) [25], singular consistency method [28], and singular robustness algorithm [29]. However, most of these singularity avoidance algorithms are analyzed based on velocity and generally determine whether a singularity occurs during motion by computing the rank of the Jacobian matrix, while the position-based singular avoidance method in the case of deriving redundant robot inverse kinematics analytic solutions has not been discussed. That is, the position of the end-effector and the position of the joint angle are used to determine whether the manipulator is in a singular configuration.

The optimum kinematic design for a 7-DOF SRS configuration serial manipulator was proposed by Hollerbach [30], Due to its structural advantages in task performance and anthropomorphic characteristics, it has been extensively studied [31,32]. There are also many studies on the kinematics and singularity of a complex space manipulator with distance offset (the axes of the spherical joints do not intersect) [33,34]. Another superior kinematic design is a 7-DOF redundant manipulator with three consecutive parallel axes, which differs from the SRS configuration series manipulator in that the axis direction of joint 3 is different. There is little research on the kinematics and singularity of such a three consecutive parallel axes 7-DOF redundant manipulator.

In this paper, the kinematics and singularities of a typical 7-DOF manipulator with three consecutive parallel axes without distance offset are researched. The auxiliary parameters of its inverse kinematics are determined by self-motion analysis, and the analytical solution of its inverse kinematics is obtained. The singularity of this manipulator and the singular avoidance are studied.

## 2. Kinematics

### 2.1. Forward Kinematics

The robot that is researched is composed of seven revolute joints with three consecutive parallel axes. To realize kinematic control and describe the kinematic relationship between the pose of the robot’s end-effector and joint angle, the mechanical structure and joint coordinate system of this manipulator are shown in Figure 1.

The base coordinate frame is fixed to the ground, and each joint coordinate frame is defined based on D–H rules. To simplify the mathematical model, the tool coordinate system is not considered. The D–H parameters of the manipulator are presented in Table 1, in which the length of each link is *d*_1_ = |***OS***|, *d*_2_ = |***SA***|, *d*_3_ = |***AE***|, *d*_4_ = |***EW***|.

According to the D–H rules and parameters of this manipulator listed in Table 1, the coordinate transform matrix of adjacent joints can be obtained by using a homogeneous transformation matrix Tii−1.
(1)Tii−1=[cθi−sθi0ai−1sθicαi−1cθicαi−1−sαi−1−disαi−1sθisαi−1cθisαi−1cαi−1dicαi−10001]
where $c\theta_{i}=\cos(\theta_{i})$, $s\theta_{i}=\sin(\theta_{i})$, $c\alpha_{i-1}=\cos(\alpha_{i-1})$, $\;s\alpha_{i-1}=\sin(\alpha_{i-1})$.

When the values of all joint angles are known, the homogeneous transformation matrix of the end-effector relative to the base can be obtained by multiplying seven homogeneous transformation matrices.
(2)T70=T10T21T32T43T54T65T76

According to Equation (2) the forward kinematic solution can be obtained, which indicates the position and orientation of the end-effector.

### 2.2. Self-Motion Analysis

According to the mechanical characteristics of the manipulator, the geometric simplification of the manipulator is carried out. Since the axes of the last three joints of the manipulator intersect at a point, when the axes are not coplanar, the rotation of the axes of the last three joints corresponds to a spherical pair centered on *W*, as shown in Figure 2.

When the position of the end-effector is determined, the location of *W* is known. Since the axes of the joints 2, 3, 4 are parallel to each other and the locations of point *O* and point *S* remain unchanged, it can be known that point *O*, *S*, *A*, *E*, and *W* are on the same plane *P* and the possible motion of the manipulator is the movement of the links *SA*, *AE*, and *EW* on the plane *P*. It can be considered as the self-motion of the 7-DOF manipulator. The movement can be seen as the planar four-bar motion with SW as the fixed link. The positions of points *A* and *E* are determined by only one parameter. The angle *φ* between *SA* and *SW* is considered as the parameter of the self-motion, as shown in Figure 3.

When the end-effector of the manipulator is at a different position, the length of *SW* is different. The variation range of redundant angle *φ* can be derived according to the kinematic principle of planar four-bar mechanism, which lays a foundation for solving the inverse kinematics of the redundant manipulator based on the redundant angle *φ*.

### 2.3. Inverse Kinematics

According to the configuration characteristics of this manipulator, the last three joints of the manipulator are equivalent to a spherical joint, which can reach any orientation. Therefore, for the inverse kinematics, the separation of position and orientation can be achieved. In this way, the first four joints of the manipulator determine the position of the end-effector; the last three joints determine the orientation of the end-effector.

Assuming the position and orientation of the manipulator are known as
(3)T70=[R70P7001]=[nxoxaxpxnyoyaypynzozazpz0001]

The inverse kinematics can be solved by the following 5 steps:*θ*_1_

As shown in Figure 4, since the axes of the joints 2, 3, 4 are parallel to each other, the position of the self-motion plane *P* is only determined by joint 1. Therefore, the expression of *θ*_1_ is
(4)$$\theta_{1}=\mathrm{atan}2\left(n_{0}\times n_{1}\cdot z_{1},n_{0}\cdot n_{1}\right)+\frac{\left(1-GK_{1}\right)\pi }{2}$$
where *GK*_1_= ±1 corresponding to the two solutions of *θ*_1_, ***n***_0_ and ***z***_1_ is the axis direction of the joint 1 in the base coordinate system, ***n***_0_ and ***n***_1_ are the normal vectors of the initial principal plane *P*’ and the corresponding principal plane *P* of the current point *W*.
(5)$$\begin{array}{c} n_{0}={\left[\begin{array}{ccc} 0 & 1 & 0 \end{array}\right]}^{T}\\ n_{1}=OS\times SW={\left[\begin{array}{ccc} -d_{1}p_{y} & d_{1}p_{x} & 0 \end{array}\right]}^{T} \end{array}$$
where
(6)$$\begin{array}{l} OS={\left[\begin{array}{ccc} 0 & 0 & d_{1} \end{array}\right]}^{T}\\ SW=OW-OS={\left[\begin{array}{ccc} p_{x} & p_{y} & p_{z}-d_{1} \end{array}\right]}^{T} \end{array}$$


2.
*θ*
_2_



As shown in Figure 5, *θ*_2_ can represent the angle of rotation from ***x***_1_ to ***x***_2_ around ***z***_2_, and ***x***_2_ is collinear with the link *SA*. If the redundancy angle *φ* is known, *θ*_2_ can be expressed as
(7)$$\theta_{2}=GK_{1}\cdot \varphi +\varphi_{SM}$$
where *φ* is the redundant angle, $\varphi_{SM}$ can represent the angle of rotation from ***x***_1_ to *SW* around ***z***_2_.

According to *θ*_1_, $\varphi_{SM}$ and ***z***_2_ can be obtained.
(8)$$\begin{array}{c} \;\varphi_{SM}=\mathrm{atan}2(x_{1}\times SW\cdot z_{2},x_{1}\cdot SW)\\ z_{2}=-n_{1} \end{array}$$

The unit vectors ***z***_2_ and ***x***_1_ represent the *z*-axis vector of the joint axis 2 and the *x*-axis vector of the joint axis 1, respectively.


3.
*θ*
_3_



The position of point *A* can be solved when the angles *θ*_1_ and *θ*_2_ are known. At this time, *AW*, *AE*, and *EW* form a fixed triangle with a fixed edge length in Figure 6. In this way, the position of point *E* can be obtained.
(9)$$\begin{array}{c} \;\psi =GK_{2}\cdot \arccos(\frac{d_{3}^{2}+{\left|AW\right|}^{2}-d_{4}^{2}}{2d_{3}\left|AW\right|})\\ AE=d_{3}\cdot \mathrm{Rot}(z_{2},\psi )\frac{AW}{\left|AW\right|} \end{array}$$
where *GK*_2_= ±1 corresponds to the two possible positions of the point *E*, corresponding to point *E* above and point *E*’ below, respectively. |***AW***| represents the distance from point *A* to *W*.

*θ*_3_ is the angle of rotation from *SA* to *AE* around ***z***_3_, so *θ*_3_ can be expressed as follows:(10)$$\;\theta_{3}=\mathrm{atan}2(SA\times AE\cdot z_{3},SA\cdot AE)$$


4.
*θ*
_4_



According to the D–H method, *θ*_4_ is the angle of rotation from ***x***_3_ to ***x***_4_ around ***z***_4_.
(11)$$\;\theta_{4}=\mathrm{atan}2(x_{3}\times x_{4}\cdot z_{4},x_{3}\cdot x_{4})$$
where ***z***_2_ is parallel to ***z***_4_. ***X***_3_ can be obtained by coordinate transformation according to *θ*_1_, *θ*_2,_, and *θ*_3_. Because *EW* and ***z***_5_ are collinear with each other, ***x***_4_ can be expressed as
(12)$$x_{4}=z_{4}\times z_{5}=z_{3}\times \frac{EW}{\left|EW\right|}$$


5.
*θ*
_5,_
*θ*
_6,_
*θ*
_7_



According to the above analysis, the values of the first four joints have been obtained, and the transformation matrix from the initial orientation to the target orientation can be described as
(13)R74=R40−1⋅R70=[nx′ox′ax′ny′oy′ay′nz′oz′az′]
where R40 is the orientation matrix of joint coordinate system {4} in the base coordinate system, which is calculated by *θ*_1_, *θ*_2_, *θ*_3_, and *θ*_4_. R70 is known and given by Equation (3).

According to the coordinate transformation of the D–H method, R74 is the orientation matrix of joint coordinate system {7} in coordinate system {4}, which can also be described as
(14)R74=R54R65R76=[c5c6c7−s5s7−c5s7−c6c7s5c5s6c7s6−s7s6−c6c5s7+c6c7s5c5c7−s5c6s7s5s6]
where *s* and *c* are the abbreviations of sine function and cosine function, respectively, and the subscript number *i* (*i* = 1, …, 7) is the corresponding joint angle *θ_i_*. For example, *c*_6_ = cos(*θ*_6_).

When $a_{y}^{\prime }=-1$, that is $\theta_{6}=0,\pm \pi$. The axes of joint 5 and joint 7 of the manipulator coincide; at this time, only the sum of values of the joint 5 and the joint 7 can be obtained, and the joint angles can be arbitrarily selected as long as the sum is unchanged.
(15)$$\theta_{5}+\theta_{7}=\mathrm{atan}2\left(n_{z}^{\prime },n_{x}^{\prime }\right)$$

When $a_{y}^{\prime }\neq -1$, that is $s_{6}\neq 0$. Due to the existence of the cosine function, there are two solutions. Introducing variables *GK*_3_= ±1 reflect this position
(16)$$\begin{array}{l} \theta_{6}=GK_{3}\cdot \arccos(-a_{y}^{\prime })\\ \theta_{5}=\mathrm{atan}2(\frac{a_{z}^{\prime }}{s_{6}},\frac{a_{x}^{\prime }}{s_{6}})\\ \theta_{7}=\mathrm{atan}2(\frac{-o_{y}^{\prime }}{s_{6}},\frac{n_{y}^{\prime }}{s_{6}}) \end{array}$$

In summary, eight discrete inverse solutions of the manipulator can be obtained with given position-orientation ***T*** and redundant angle *φ*, and the unique inverse solutions of the control variable *GK*_1_, *GK*_2_, and *GK*_3_ can be obtained.

## 3. Singularity Analysis

The Jacobian matrix reflects the mapping relationships between the generalized velocity (linear velocity and angular velocity) of the end-effector and the joint velocities. The singularity condition of the manipulator can be obtained by analyzing the conditions when the determinant of the Jacobian matrix is equal to zero. However, since the Jacobian matrix of the redundant manipulator is not a square matrix, it is not feasible to obtain singular conditions by means of the matrix determinant. The method based on the elementary transformation [26] of the Jacobian matrix is widely used. That is, the Jacobian matrix is simplified by the elementary transformation. The singular conditions are analyzed and concentrated in a low-dimensional submatrix by the form of the block matrix, and then the singularity conditions of the submatrix are analyzed to obtain the singularity conditions of the entire Jacobian matrix [23].

### 3.1. Jacobian Matrix

In this paper, the vector cross-product method [35] is used to establish the Jacobian matrix of the manipulator. The relationship between the terminal velocity and the joint velocity corresponding to a single joint is as follows:(17)$$\left[\begin{array}{c} \omega \\ v \end{array}\right]=J(\theta )\dot{\theta }$$
where ***ω*** and ***v*** are the angular velocity and linear velocity of the end effector, respectively. $\dot{\theta }$ is the joint velocity vector and ***J***(***θ***) is the Jacobian matrix. Since the seven joints of this manipulator are all revolute joints, the *i*th column of the Jacobian matrix can be expressed as:(18)Ji=[zizi×p75i]
where ***z****_i_* is the unit vector of the *i*th joint axis, p75i is the representation of the origin of the joint coordinate system {7} in the joint coordinate system {5} relative to the position vector in the coordinate system {*i*}.

Adopting the above method, the Jacobian matrix of the manipulator is established with joint 5 as the reference coordinate system as follows:(19)J5=[c5s234s5s5s500s6−s5s234c5c5c5010−c23400010c6J41J42J43−d4c5000J51J52J53d4s50000J62d3c40000]
where the subscript of the symbol represents the sine or cosine of the corresponding joint variable sum, such as $s_{234}=\sin(\theta_{2}+\theta_{3}+\theta_{4})$.
$$\begin{array}{l} J_{41}=d_{4}s_{234}s_{5}+d_{3}c_{23}s_{5}+d_{2}c_{2}s_{5}\\ J_{51}=d_{4}s_{234}c_{5}+d_{3}c_{23}c_{5}+d_{2}c_{2}c_{5}\\ J_{42}=-d_{4}c_{5}-d_{3}s_{4}c_{5}-d_{2}s_{34}c_{5}\\ J_{52}=d_{4}s_{5}+d_{3}s_{4}s_{5}+d_{2}s_{34}s_{5}\\ J_{62}=d_{3}c_{4}+d_{2}c_{34}\\ J_{43}=-d_{4}c_{5}-d_{3}s_{4}c_{5}\\ J_{53}=d_{4}s_{5}+d_{3}s_{4}s_{5} \end{array}$$

Because of the wrist structure of the manipulator, the last three columns of the Jacobian matrix are in the simplest form. In this way, the Jacobian matrix can be separated into a low-dimensional matrix by elementary transformation. In the following section, the expression of the low-dimensional matrix is expressed, and the singularity conditions of the Jacobian matrix are obtained by analyzing the singularity conditions of the matrix.

### 3.2. Singularity Conditions

According to Equation (19), the lower right corner of the Jacobian matrix is simplified to a 3 × 3 zero matrix, which can be discussed by dividing the Jacobian matrix into blocks. The following discussion is based on the value of *S*_6_:*S*_6_ = 0

When *S*_6_ = 0, the Jacobian matrix can be transformed into the following form
(20)J5s6=0=[λ1000λ2010λ3101λ4000λ5000λ6000]
where *λ*_1_ to *λ*_6_ in Equation (20) are sub-matrices of the Jacobian matrix to make the structure of the Jacobian matrix look more concise.

To facilitate the singularity analysis, elementary transformation is performed on the matrix in Equation (20), and the matrix is written into the form of a block matrix.
(21)J⌢5s6=0=[λ3101λ2010λ1000λ4000λ5000λ6000]=[S115S125S215O4×3]
where $O_{4\times 3}$ is a 4 × 3 zero matrix, and the other sub-matrices are as follows:(22)S115=[λ3λ2]S125=[101010]S215=[λ1λ4λ5λ6]

According to the characteristics of the matrix rank, S125 is a row full rank matrix, i.e., rank(S125)=2. So, the rank of the block-triangle matrix can be computed according to the following equation [36]:(23)rank(J⌢5s6=0)=rank([S115S125S215O4×3])=rank(S125)+rank(S215)

If J⌢5s6=0 is singular, it should satisfy:(24)rank(J⌢5s6=0)<6

Namely,
(25)rank(S215)<4

We only need to analyze the singularity S215.
(26)S215=[c5s234s5s5s5A21A22A23−d4c5A31A32A33d4s50A42d3c40]
where
$$\begin{array}{l} A_{21}=s_{5}(d_{4}s_{234}+d_{3}c_{23}+d_{2}c_{2})\\ A_{22}=-c_{5}(d_{4}+d_{3}s_{4}+d_{2}s_{34})\\ A_{23}=-c_{5}(d_{4}+d_{3}s_{4})\\ A_{31}=c_{5}(d_{4}s_{234}+d_{3}c_{23}+d_{2}c_{2})\\ A_{32}=s_{5}(d_{4}+d_{3}s_{4}+d_{2}s_{34})\\ A_{33}=s_{5}(d_{4}+d_{3}s_{4})\\ A_{42}=d_{3}c_{4}+d_{2}c_{34} \end{array}$$

Since S215 is a square matrix, the singularity conditions of the Jacobian matrix can be obtained by judging whether the determinant of the matrix is equal to zero.
(27)det(S215)=−d2d3s3s5(d2c2+d3c23+d4s234)

Then, it can be obtained that, when det(S215)=0, the rank of the Jacobian matrix is less than 6; that is, it is in the singular position. For det(S215)=0, there are three cases: $d_{2}c_{2}+d_{3}c_{23}+d_{4}s_{234}=0$, $s_{3}=0$, and $s_{5}=0$.


2.*S*_6_ ≠ 0


When *S*_6_ ≠ 0, the Jacobian matrix can be expressed in the form of block matrix
(28)J5s6≠0=[λ100s6λ2010λ310c6λ4000λ5000λ6000]=[S115S125S215O3×3]
where S125 is a square matrix and its determinant value is det(S125)=−s6. Because of *S*_6_ ≠ 0, to make Jacobian matrix singular, it needs to satisfy
(29)rank(S215)<3

The singularity of the Jacobian matrix can be judged by analyzing the singularity of S215. The expression of S215 is as follows:(30)S215=[A11A12A13−d4c5A21A22A23d4s50A32d3c40]
where
$$\begin{array}{l} A_{11}=(d_{4}s_{234}+d_{3}c_{23}+d_{2}c_{2})s_{5}\\ A_{12}=-(d_{4}+d_{3}s_{4}+d_{2}s_{34})c_{5}\\ A_{13}=-(d_{4}+d_{3}s_{4})c_{5}\\ A_{21}=(d_{4}s_{234}+d_{3}c_{23}+d_{2}c_{2})c_{5}\\ A_{22}=(d_{4}+d_{3}s_{4}+d_{2}s_{34})s_{5}\\ A_{23}=(d_{4}+d_{3}s_{4})s_{5}\\ A_{32}=d_{3}c_{4}+d_{2}c_{34} \end{array}$$

Since the matrix S215 is a rectangular matrix, the determinant method cannot obtain the condition of matrix rank deficiency. Therefore, it is necessary to be discussed according to the structural characteristics of the matrix. By looking at the first column of matrix S215, we can see the following:If $A=d_{2}c_{2}+d_{3}c_{23}+d_{4}s_{234}=0$, then *A*_11_ = *A*_21_ = 0, and S215 becomes a square matrix.



(31)
S21A=05=[−(d4+d3s4+d2s34)c5−(d4+d3s4)c5−d4c5(d4+d3s4+d2s34)s5(d4+d3s4)s5d4s5d3c4+d2c34d3c40]



Since det(S21A=05)=0, *S*_6_ ≠ 0 and $d_{2}c_{2}+d_{3}c_{23}+d_{4}s_{234}=0$ is one of the conditions of matrix rank deficiency. Consider Equation (27), *S*_6_ = 0 and $d_{2}c_{2}+d_{3}c_{23}+d_{4}s_{234}=0$ is also one of the conditions of matrix rank deficiency. Therefore, $d_{2}c_{2}+d_{3}c_{23}+d_{4}s_{234}=0$ is one of the singular conditions of the matrix regardless of whether *S*_6_ is equal to 0 or not.


If $A=d_{2}c_{2}+d_{3}c_{23}+d_{4}s_{234}\neq 0$
, S215 is transformed into an elementary transformation:



(32)
S21A≠05=[10000(d4+d3s4+d2s34)s5/A(d4+d3s4)s5/Ad4s5/A0d3c4+d2c34d3c40]


If *S*_5_ = 0, then the matrix can be written in the following form after the elementary transformation:(33)S21A≠0, s5=05=[10000(d4+d3s4+d2s34)c5/A(d4+d3s4)c5/Ad4c5/A0d3c4+d2c34d3c40]

It can be seen that the value of joint 5 does not affect the rank of the matrix, so the only condition in which the rank of the matrix S215 is not full is
(34)$$c_{34}=0\;,\;c_{4}=0$$

In summary, there are four singularity conditions for the 7-DOF redundant manipulator, as shown in Table 2. If the joint angles of the manipulator satisfy any of them, the manipulator will be in a singular configuration.

## 4. Singularity Avoidance

It can be seen from the above analysis that the singularity conditions are described in the joint space. However, in practice, the end-effector of this manipulator is usually planned, and the singularity configuration is impossible to be predicted and avoided in motion planning because of the infinite number of the inverse kinematics solutions of the redundant manipulator. In the following section, the above-mentioned singular configuration is described based on position, and a singularity avoidance method is proposed for the avoidable singularities.

### 4.1. Type I Singularity

According to the expression analysis of condition 1 and condition 2, there are four possible configurations of manipulators regarding condition 1, which are shown in Table 3, and the configurations regarding condition 2 are shown in Table 4.

The first two of the four singularity conditions described above are all related to joint 6, so both condition 1 and condition 2 can be regarded as the first type of singularity. Since both condition 1 and condition 2 contain *S*_6_ = 0, such singular problems can be solved by avoiding the configuration corresponding to *S*_6_ = 0 through self-motion.

The following is the analysis of the characteristics of the manipulator when *S*_6_ = 0. When *S*_6_ = 0, the rotation axes of the joints 5 and 7 coincide with each other, and, since the axes of joints 2, 3, and 4 are parallel to each other, the axes of joints 5 and 7 must also be in the self-motion plane *P* formed by joints 2, 3, and 4, as shown in Figure 7.

Since the axis of the last joint 7 lies on the main plane *P*, according to the Equations (3) and (5), it can be expressed as
(35)$$n_{1}\cdot a=0$$
where $a={\left[\begin{array}{ccc} a_{x} & a_{y} & a_{z} \end{array}\right]}^{T}$, ***n***_1_ is the normal vector to the plane *P*. Simplified Equation (35) can be obtained:(36)$$p_{x}a_{y}-p_{y}a_{x}=0$$

When the singularity occurs in Table 3, according to the geometric theorem forming the triangle, the distance *SW* between the shoulder point and wrist point of the manipulator and the corresponding redundancy angle meet the following equation:(37)$$\varphi =\{\begin{array}{cc} \pm \varphi_{1}\;, & \left|d_{2}+d_{3}-d_{4}\right|<\left|SW\right|<d_{2}+d_{3}+d_{4}\\ \pm \varphi_{2}\;, & \left|\left|d_{2}-d_{3}\right|-d_{4}\right|<\left|SW\right|<\left|d_{2}-d_{3}\right|+d_{4} \end{array}\;$$
where the angle ± *φ*_1_ represents the corresponding redundancy angle when singular conditions of *θ*_3_ = 0, *θ*_6_ = 0, or π are satisfied, as shown in Figure 7a. In this case, angle *φ*_1_ can be expressed as the following equation:(38)$$\varphi_{1}=\arccos\frac{{\left(d_{2}+d_{3}\right)}^{2}+{\left|SW\right|}^{2}-d_{4}^{2}}{2(d_{2}+d_{3})\left|SW\right|}$$
where the distance of *SW* can calculated from a given end-effector position, i.e., Equation (6). At this point, the axis vector of joint 5 (i.e., vector ***EW*** or ***E*’*W***) can be expressed as:(39)$$\begin{array}{l} SE=Rot(z_{2},\varphi_{1})\cdot \frac{SW}{\left|SW\right|}\cdot SE\\ EW=SW-SE\\ SE’=Rot(z_{2},-\varphi_{1})\cdot \frac{SW}{\left|SW\right|}\cdot SE\\ E’W=SW-SE’ \end{array}$$
where $Rot(z_{2},\pm \varphi_{1})$ represents the rotation matrix for the rotation of angle ± *φ*_1_ about the ***z***_2_ axis. The angle ± *φ*_2_ represents the corresponding redundancy angle when singular condition of *θ*_3_ = π, *θ*_6_ = 0, or π are satisfied, as shown in Figure 7b. In this case, angle *φ*_2_ can be expressed as the following equation:(40)$$\varphi_{2}=\arccos\frac{{\left(d_{2}-d_{3}\right)}^{2}+{\left|SW\right|}^{2}-d_{4}^{2}}{2\left|d_{2}-d_{3}\right|\cdot \left|SW\right|}$$

So, the axis vector of joint 5 can be expressed as:(41)$$\begin{array}{l} SE=Rot(z_{2},\varphi_{2})\cdot \frac{SW}{\left|SW\right|}\cdot SE\\ EW=SW-SE\\ SE’=Rot(z_{2},-\varphi_{2})\cdot \frac{SW}{\left|SW\right|}\cdot SE\\ E’W=SW-SE’ \end{array}$$

When the axis of joint 7 is collinear with *EW* or *E’W*, the following equation is satisfied:(42)$$\frac{EW}{\left|EW\right|}\cdot a=\pm 1\;\mathrm{or}\;\frac{E’W}{\left|E’W\right|}\cdot a=\pm 1\;$$

Therefore, when the position and orientation of the end-effector are given, the relationship between ***EW*** vector and the axis vector ***a*** of joint 7 can be determined to know whether the manipulator is in the singular pose corresponding to singular condition 1.

In order to avoid singular configuration corresponding to singular condition 1, the redundant angle *φ* obtained according to Equation (38) and Equation (40) should be avoided during motion planning. However, singular condition 2 is only related to joint 5 and joint 6, and this singular type cannot be determined by the given end-effector position and orientation.

### 4.2. Type II Singularity

The last two of the four singularity conditions are related to joints 2, 3, and 4, so both conditions 3 and 4 can be regarded as the second kind of singularity. Since the first four joints determine the position of the end-effector, they can be regarded as the position singularity. From the perspective of workspace, position singularity can be divided into boundary singularity and internal singularity. Boundary singularity refers to the singularity caused by the position of the end-effector at the boundary of the workspace. Internal singularity refers to the position of the end-effector in the workspace when it is in the singular configuration. Whether it can be avoided needs to be discussed.

According to the geometric projection relationship of Figure 8, the projection of point *W* in the horizontal direction *x* is shown in Equation (43).
(43)$$L_{x}=d_{2}c_{2}+d_{3}c_{23}+d_{4}c_{\beta }$$
where $\beta =\theta_{234}-\pi /2$. When this condition $L_{x}=0$ is satisfied, that is, singular condition 3, its geometric meaning is that the projection of point *W* in the horizontal direction *x* is zero. In other words, the end-effector of the manipulator is located on the axis of joint 1. At the moment, the position of the end-effector of the manipulator satisfies
(44)$$\;p_{x}=p_{y}=0$$

From the analysis of the working space, the end-effector of the manipulator loses the movement perpendicular to the normal direction of the main plane, so the axis of joint 1 is also the boundary of the working space and cannot be avoided, as shown in Figure 9.

According to the expression analysis of condition 4, there are four possible joint combinations of joint 3 and joint 4, as shown in Table 5.

When *θ*_3_ = 0, *θ*_4_ = π/2, the manipulator is in the singularity position of the boundary of the workspace, which is an unavoidable singularity, and the other three are internal singularities. These four configurations have common motion characteristics. In these cases, points *S* and *W* are on a straight line; the length of *SW* is as follows:(45)$$\left|SW\right|=\{\begin{array}{ll} d_{2}+d_{3}+d_{4}\;, & \theta_{3}=0,\;\theta_{4}=\pi /2\\ d_{2}+d_{3}-d_{4}\;, & \theta_{3}=0,\;\theta_{4}=-\pi /2\\ d_{2}-d_{3}+d_{4}\;, & \theta_{3}=\pi ,\;\theta_{4}=\pi /2\\ d_{2}-d_{3}-d_{4}\;, & \theta_{3}=\pi ,\;\theta_{4}=-\pi /2 \end{array}$$

At this time, the corresponding redundancy angle is *φ* = 0. Therefore, it can be concluded that, when the end-effector of the manipulator is in the avoidable singularity, the control variable *φ* is unequal to zero to avoid the current singularity configuration.

## 5. Simulation and Discussion

In this section, some simulation examples will be given to validate the previous analysis. Assume that the specific dimensions of the manipulator are as follows:$$d_{1}=10\mathrm{mm},\;d_{2}=20\mathrm{mm},\;d_{3}=30\mathrm{mm},\;d_{4}=20\mathrm{mm}$$

### 5.1. Case 1

Assuming that the initial position and orientation of the end-effector are
(46)$$X_{e0}=\left[30\mathrm{mm},\;30\mathrm{mm},\;30\mathrm{mm},\;0.6\mathrm{rad},\;0.4\mathrm{rad},\;1.2\mathrm{rad}\right]$$

The terminal position and orientation of the end-effector are
(47)$$X_{\mathrm{ef}}=\left[40\mathrm{mm},\;10\mathrm{mm},\;10\mathrm{mm},\;0\mathrm{rad},\;0\mathrm{rad},\;0\mathrm{rad}\right]$$

When the manipulator is performing a task, intuitively, from an energy-saving point of view, it should try to move more joints near the end-effector of the robot arm and less joints near the base of the manipulator. Suppose we are now given the optimization requirement to keep joint 2 as motionless as possible during the motion and the motion of the remaining joints according to the principle of “the shortest trip”, that is, the amount of motion of each joint is minimized; the possible solution with the smallest joint angle difference from the previous time is selected to ensure a smooth transition of joint motion between the two times. The appropriate redundancy angle *φ* is selected to minimize the change of upper arm corresponding to adjacent path points. That is,
(48)$$\Delta \theta_{2}=\left|\theta_{2}\left(i+1\right)-\theta_{2}\left(i\right)\right|$$
where $\theta_{2}\left(i\right)$ and $\theta_{2}\left(i+1\right)$ represent the angle of joint 2 between the current time and the next time. The symbol *i* represents the *i*th waypoint among many discrete points of a trajectory.

In Figure 10, the planned trajectory can be obtained by the interpolation between $X_{e0}$ and $X_{\mathrm{ef}}$, and the actual trajectory can be obtained by forward kinematic of the joint angle in Figure 11. The planned trajectory of the end-effector of the manipulator coincides with the actual trajectory, which proves the correctness of the algorithm of inverse kinematics. In Figure 11, the value of joint 2 does not change, which proves that this algorithm can make the manipulator move along the planned trajectory while keeping the upper arm as still as possible according to the optimization requirements.

### 5.2. Case 2

For manipulators, maneuverability [37] is usually used as a measure of the dexterity of the manipulator, which can be expressed by Equation (49)
(49)$$\omega =\sqrt{\det(J\cdot J^{T})}$$

When *ω* = 0, it is in the singularity state. When *ω* > 0, it is in the non-singular state. To make the expression of singularity more obvious, Equation (49) is modified to
(50)$$\lambda =1/\left(\det(J\cdot J^{T})+1\right)$$

When the location of the manipulator is in the non-singularity configuration, *λ* is close to 0, and, in the singularity configuration, *λ* is close to 1. So, *λ* can be used to judge whether the manipulator is at the singularity condition. Therefore, we can define *λ* as a singular index to judge singularity.

#### 5.2.1. The Verification of Type I Singularity

Assuming that the position and orientation of the end-effector are
(51)$$\begin{array}{l} X_{e1}=\left[30\mathrm{mm},\;10\mathrm{mm},\;10\mathrm{mm},\;2.1803\mathrm{rad},\;0.6194\mathrm{rad},\;-0.7326\mathrm{rad}\right]\\ H_{e1}=\left[\begin{array}{cccc} 0.5256 & 0.2236 & -0.8250 & 30\\ -0.4729 & -0.6708 & -0.2750 & 10\\ -0.6149 & 0.7071 & -0.4937 & 10\\ 0 & 0 & 0 & 1 \end{array}\right] \end{array}$$
where $X_{e1}$ is the pose vector composed of three-dimensional position coordinates and RPY (roll-pitch-yaw) angle, and $H_{e1}$ is its corresponding homogeneous transformation matrix.

From Equation (51), it can be seen that the relationship between the position and the orientation of the manipulator satisfies Equation (36), and Equation (37) needs to be avoided by controlling the self-motion variables, and the value of the redundancy angle corresponding to the singularity can be obtained by the Equation (38).
(52)φ=± 0.1987 rad

Figure 12 shows the relationship between the singularity index of the manipulator and the selection of self-motion variables when the position and orientation of the end-effector are determined. Most of the values are infinitely close to zero, but there is a peak corresponding to the value *λ* between *φ* = 0 and *φ* = 0.5. At this time, according to the redundant angle corresponding to the singularity configuration obtained above, it can be proved that using the above singularity judgment method for type I, that is, judging whether singularity will occur according to the end pose relationship, and the value of redundant angle *φ* that should be avoided, can be obtained.

#### 5.2.2. The Verification of Type II Singularity

The position singularity is only related to the position of the manipulator, so the given position of the end-effector is
(53)$$\begin{array}{l} X_{e2}=\left[30\mathrm{mm},\;0\mathrm{mm},\;10\mathrm{mm},\;0.2\mathrm{rad},\;0.3\mathrm{rad},\;0.5\mathrm{rad}\right]\\ H_{e2}=\left[\begin{array}{cccc} 0.8384 & -0.4183 & 0.3494 & 30\\ 0.4580 & 0.8882 & -0.0355 & 0\\ -0.2955 & 0.1898 & 0.9363 & 10\\ 0 & 0 & 0 & 1 \end{array}\right] \end{array}$$
where $X_{e2}$ is the pose vector composed of three-dimensional position coordinates and RPY angle, and $H_{e2}$ is its corresponding homogeneous transformation matrix. It can be seen from Equation (53) that the distance between *S* and *W* of the manipulator satisfies the Equation (54) (i.e., $\left|SW\right|=30\;\mathrm{mm}=d_{2}+d_{3}-d_{4}$). It is necessary to avoid singularity by controlling the variable of self-motion. The value of redundancy angle corresponding to singularity can be obtained as shown below.
(54) φ=0, π rad

Similarly, when the position and orientation of the end-effector are determined, the relationship between the singularity index of the manipulator and the self-motion variable is given. As can be seen from Figure 13, most of the singularity values are infinitely close to 0, but there is a peak corresponding to the value *λ* when the value of the redundant angle is 0 or π. At this time, according to the redundant angle corresponding to the singularity configuration obtained above, it is proved that the singularity judgment and avoidance method proposed above can avoid the type II singularity configuration.

### 5.3. Case 3

Through the previous analysis, it can be seen that this redundant manipulator can avoid singularities by self-motion. In addition, the description of singularity from the aspects of terminal posture and self-motion lays a foundation for trajectory planning. The following is a simulation and planning of the trajectory of the manipulator to verify the accuracy of the singularity avoidance.

Assuming the initial point of the trajectory of the manipulator is
(55)$$X_{e0}=\left[30\mathrm{mm},\;20\mathrm{mm},\;10\mathrm{mm},\;0\mathrm{ad},\;0.6\mathrm{rad},\;0.4\mathrm{rad}\right]$$

Two intermediate points on the trajectory are
(56)$$\begin{array}{c} X_{e1}=\left[30\mathrm{mm},\;10\mathrm{mm},\;10\mathrm{mm},\;2.1803\mathrm{rad},\;0.6194\mathrm{rad},\;-0.7327\mathrm{rad}\right]\\ X_{e2}=\left[30\mathrm{mm},\;0\mathrm{mm},\;10\mathrm{mm},\;0.2\mathrm{rad},\;0.3\mathrm{rad},\;0.5\mathrm{rad}\right] \end{array}$$

The terminal point is
(57)$$X_{\mathrm{ef}}=\left[30\mathrm{mm},-10\mathrm{mm},\;10\mathrm{mm},\;0.6\mathrm{rad},\;0.4\mathrm{rad},\;1.2\mathrm{rad}\right]$$
where $X_{e1}$ and $X_{e2}$ are the pose vectors of the end effector in case 2, which represent two types of singularity, respectively.

The end-effector of the manipulator moves according to the path planned by four nodes $X_{e0}$, $X_{e1}$, $X_{e2}$, and $X_{\mathrm{ef}}$. To plan conveniently, a cubic spline curve interpolation algorithm is used, and the total time is 100s. According to the above kinematic analysis, the redundant angle *φ* needs to be determined to solve the inverse kinematics. According to Equation (49), the values of *ω* corresponding to different redundant angles *φ* are shown in Figure 14.

In Figure 14, these black circles express the value of redundant angle *φ* that satisfies the Equation (49) is maximum. The red line can be obtained by the curve fitting of the value that black circles represent. The color of Figure 14 expresses the value of *ω* with this redundant angle *φ* and increases from blue to yellow. Since the selection of the redundancy angle *φ* is based on the maneuverability index *ω*, the larger the maneuverability index *ω*, the better the flexibility of the manipulator, so the area where the red curve passes is mostly yellow. The maneuverability index *ω* is shown in Figure 15.

In Figure 15, the value of *ω* according to different redundant angle *φ* is far greater than 0, and the above values are 10^4^ orders of magnitude, which can make the manipulator have the better configuration and flexibility when the position and orientation end-effector are satisfied.

The joint angular velocities of the manipulator during this motion are shown in Figure 16. As can be seen from the above figure, the angular velocity of each joint of the manipulator varies smoothly. Therefore, this method can be used for the control of the manipulator to make sure the manipulator stays away from the singularity.

## 6. Conclusions

The inverse kinematics and singularities of the 7-DOF redundant manipulator under study are analyzed, and the self-motion of the mechanism is characterized as a planar four-bar motion, described by the redundancy angle, and an inverse kinematics analysis algorithm of the 7-DOF redundant manipulator based on the self-motion is proposed. By analyzing the Jacobi matrix of the arm, a simplified Jacobi matrix is obtained using the coordinate system selection and primary transformation of the Jacobi matrix, and the singularity conditions of the arm are derived using the block matrix. A singular configuration avoidance algorithm based on self-motion is proposed through the selection of redundant angles to avoid singular patterns. The inverse kinematics and singularity analysis of the robotic arm are verified by simulation examples, laying the foundation for the motion planning of a 7-DOF manipulator.

## Figures and Tables

**Figure 1 sensors-21-07257-f001:**
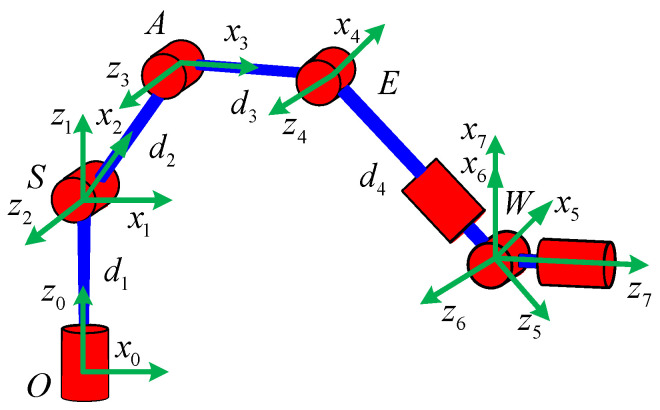
Structure of the redundant manipulator.

**Figure 2 sensors-21-07257-f002:**
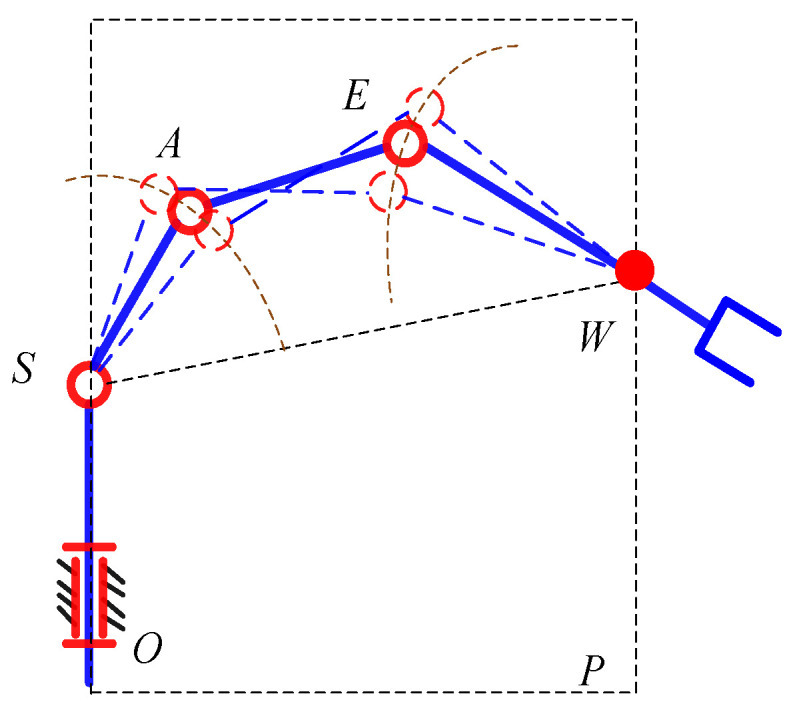
Feature of the self-motion.

**Figure 3 sensors-21-07257-f003:**
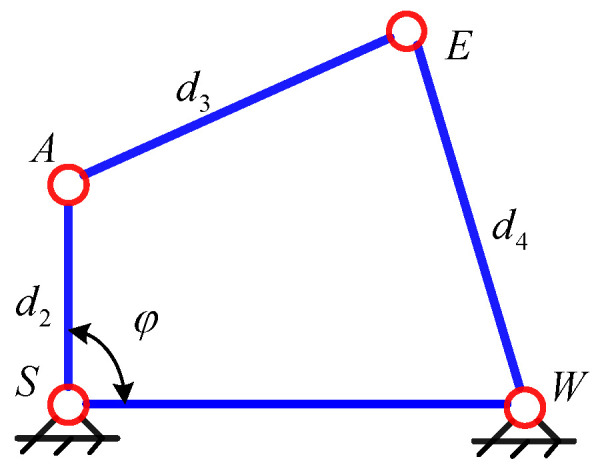
Planar four-bar mechanism.

**Figure 4 sensors-21-07257-f004:**
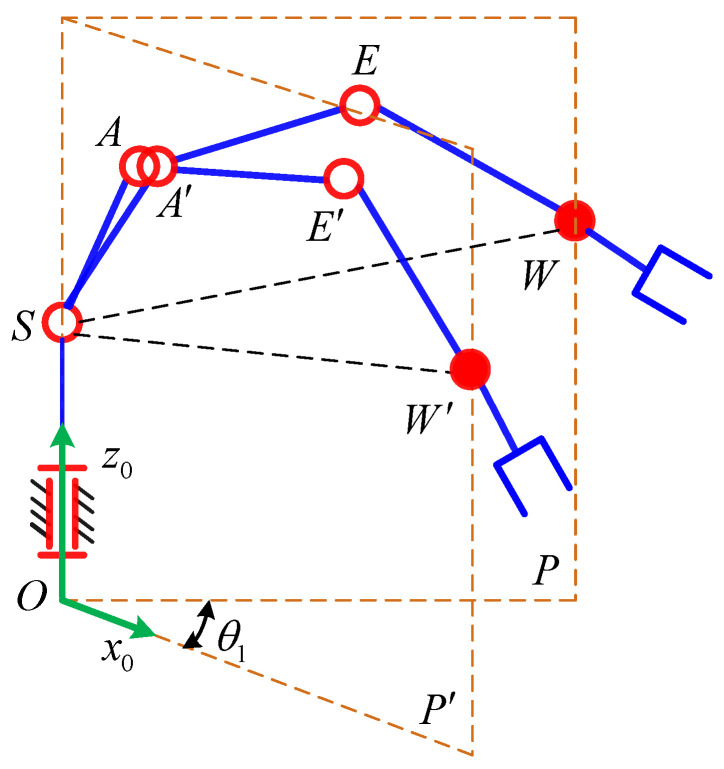
Self-motion plane.

**Figure 5 sensors-21-07257-f005:**
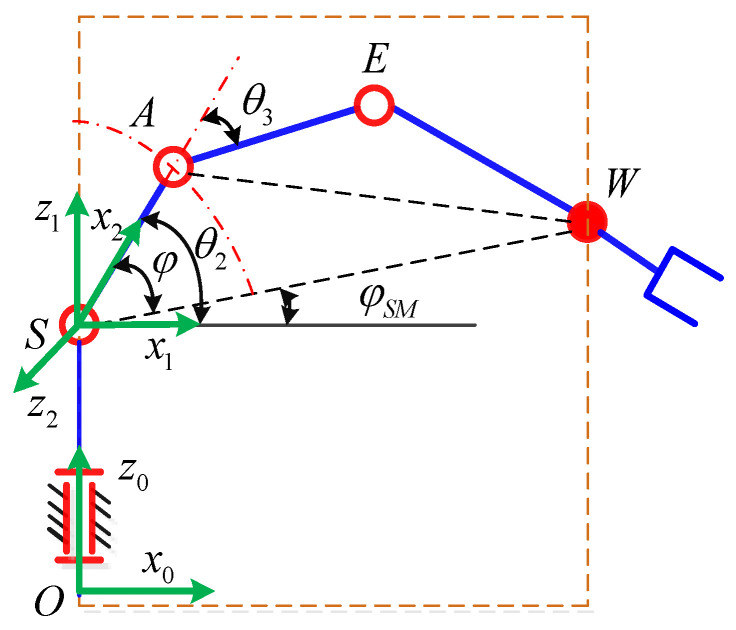
Geometric relationship of joint 2.

**Figure 6 sensors-21-07257-f006:**
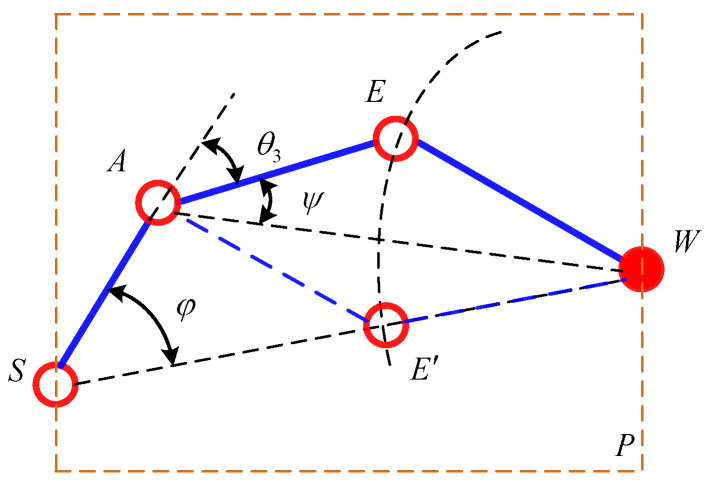
Position of the elbow E.

**Figure 7 sensors-21-07257-f007:**
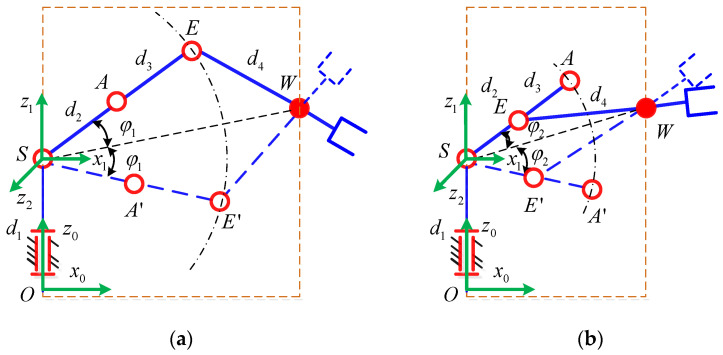
(**a**) Singular configuration corresponding to *θ*_6_ = 0 or π, *θ*_3_ = 0; (**b**) Singular configuration corresponding to *θ*_6_ = 0 or π, *θ*_3_ = π.

**Figure 8 sensors-21-07257-f008:**
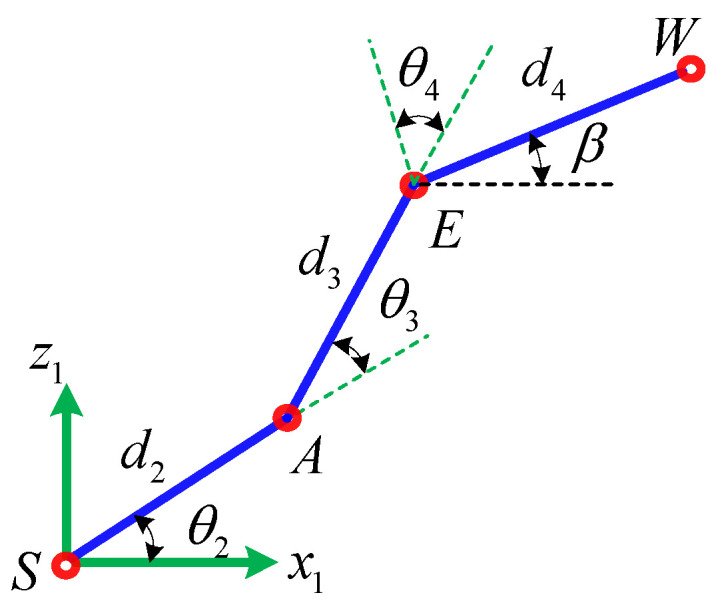
Geometric illustration of singular condition 3.

**Figure 9 sensors-21-07257-f009:**
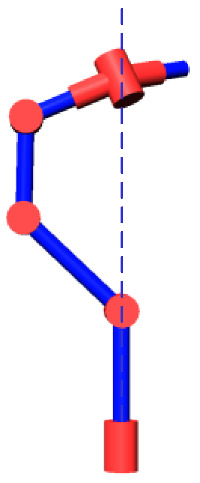
The configuration corresponding to condition 3.

**Figure 10 sensors-21-07257-f010:**
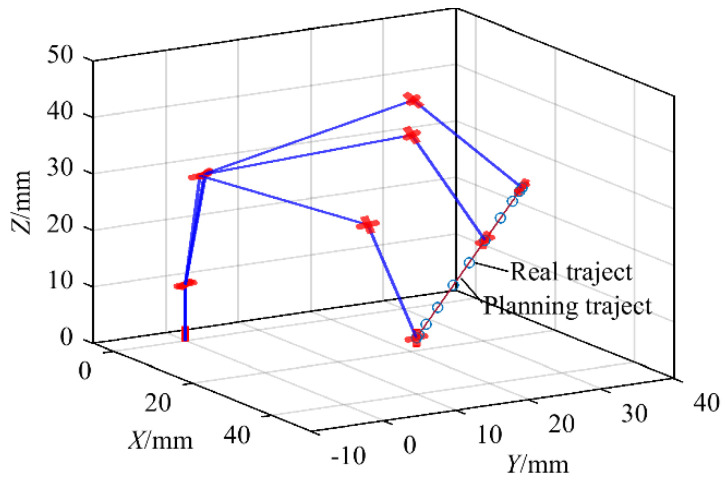
The comparison of planned trajectory and actual trajectory.

**Figure 11 sensors-21-07257-f011:**
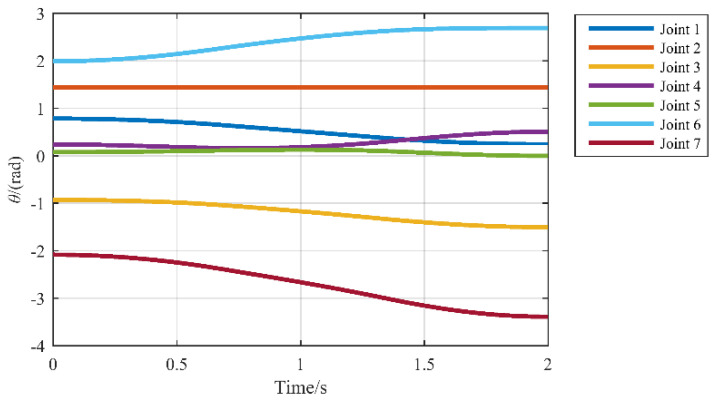
Variation of all joint angles.

**Figure 12 sensors-21-07257-f012:**
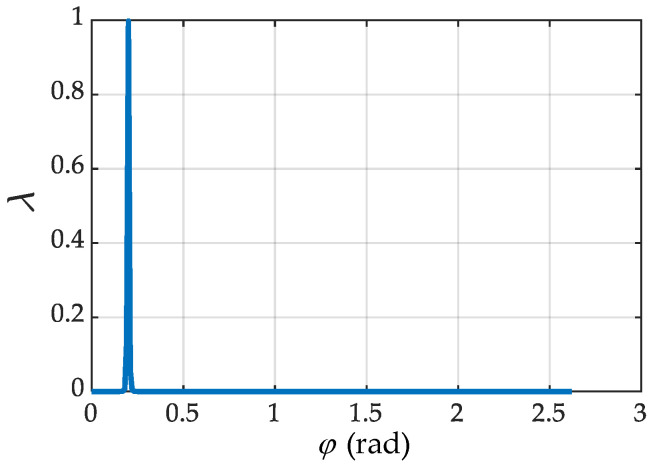
Type I.

**Figure 13 sensors-21-07257-f013:**
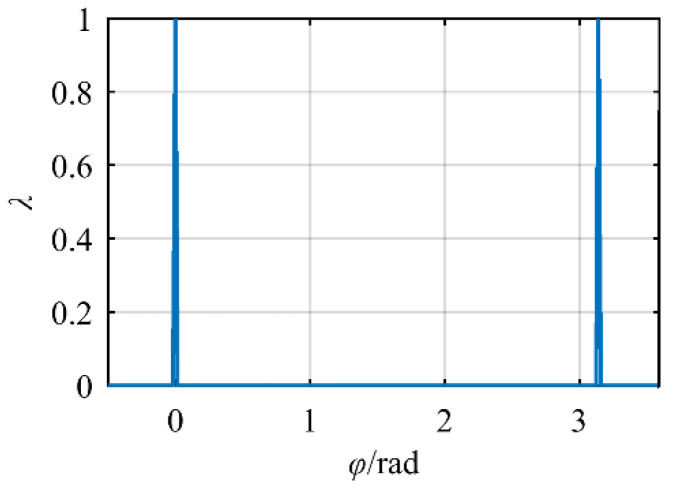
Type II.

**Figure 14 sensors-21-07257-f014:**
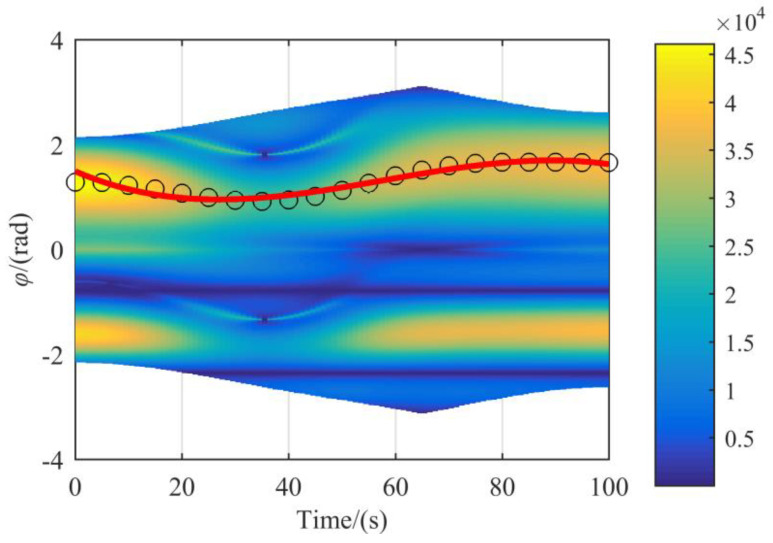
The relationship of *φ* and maneuverability index *ω* corresponding to the trajectory.

**Figure 15 sensors-21-07257-f015:**
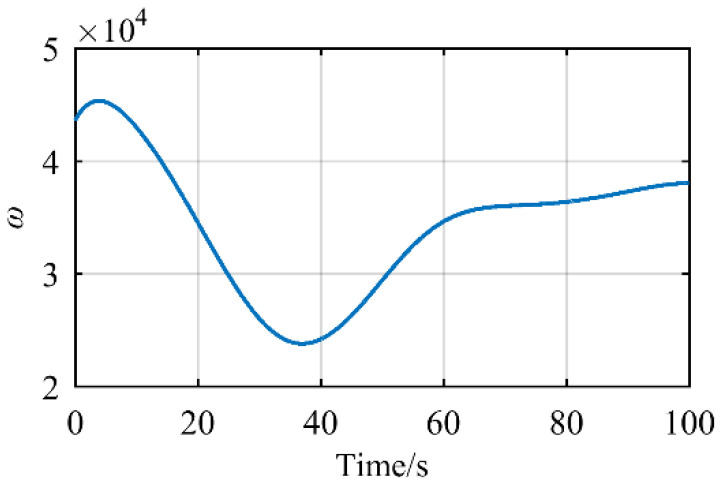
The variety of maneuverability index *ω*.

**Figure 16 sensors-21-07257-f016:**
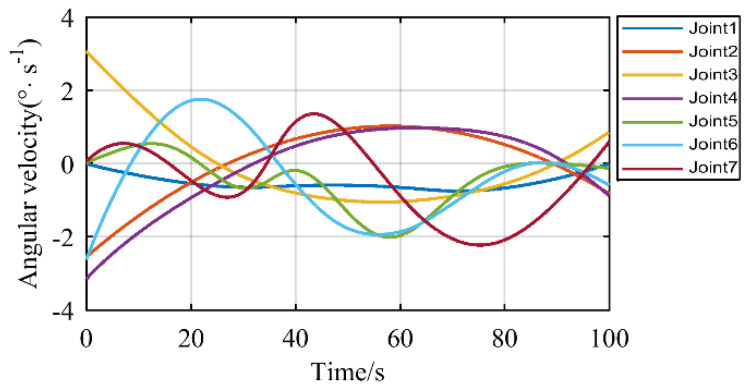
Angular velocity of each joint.

**Table 1 sensors-21-07257-t001:** D–H parameters of a 7-DOF manipulator.

Joint	*θ_i_*	$a_{i-1}$	$\alpha_{i-1}(\mathrm{rad})$	$d_{i}$
1	*θ* _1_	0	0	*d* _1_
2	*θ* _2_	0	π/2	0
3	*θ* _3_	*d* _2_	0	0
4	*θ* _4_	*d* _3_	0	0
5	*θ* _5_	0	π/2	*d* _4_
6	*θ* _6_	0	π/2	0
7	*θ* _7_	0	π/2	0

**Table 2 sensors-21-07257-t002:** Joint singularity conditions.

Condition *i*	Joint
1	$s_{6}=0,\;s_{3}=0$
2	$s_{6}=0,\;s_{5}=0$
3	$d_{2}c_{2}+d_{3}c_{23}+d_{4}s_{234}=0$
4	$c_{34}=0,\;c_{4}=0$

**Table 3 sensors-21-07257-t003:** Singularities corresponding to condition 1.

Singular condition	$\begin{array}{l} \theta_{3}=0\\ \theta_{6}=0 \end{array}$	$\begin{array}{l} \theta_{3}=0\\ \theta_{6}=\pi \end{array}$	$\begin{array}{l} \theta_{3}=\pi \\ \theta_{6}=0 \end{array}$	$\begin{array}{l} \theta_{3}=\pi \\ \theta_{6}=\pi \end{array}$
Configuration characteristics	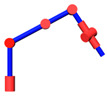	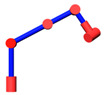	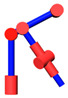	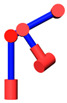

**Table 4 sensors-21-07257-t004:** Singularities corresponding to condition 2.

Singular condition	$\begin{array}{l} \theta_{5}=0\\ \theta_{6}=0 \end{array}$	$\begin{array}{l} \theta_{5}=0\\ \theta_{6}=\pi \end{array}$	$\begin{array}{l} \theta_{5}=\pi \\ \theta_{6}=0 \end{array}$	$\begin{array}{l} \theta_{5}=\pi \\ \theta_{6}=\pi \end{array}$
Configuration characteristics	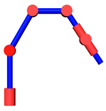	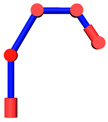	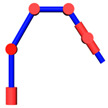	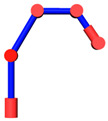

**Table 5 sensors-21-07257-t005:** Singularities corresponding to condition 4.

Singular condition	$\begin{array}{l} \theta_{3}=0\\ \theta_{4}=\pi /2 \end{array}$	$\begin{array}{l} \theta_{3}=0\\ \theta_{4}=-\pi /2 \end{array}$	$\begin{array}{l} \theta_{3}=\pi \\ \theta_{4}=\pi /2 \end{array}$	$\begin{array}{l} \theta_{3}=\pi \\ \theta_{4}=-\pi /2 \end{array}$
Configuration characteristics	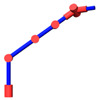	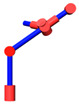	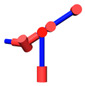	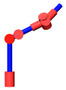

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
