# Peer review of "Kinematics and Singularity Analysis of a 7-DOF Redundant Manipulator"

_sensors, 2021, doi:10.3390/s21217257_

Round 1

Reviewer 1 Report

  1. In the abstract, it should use " Jacobian matrix" instead of Jacobi, and the " singular conditions" should be "conditions for kinematic singularities" and "the singularity conditions in the joint space" should be " the kinematic singularities in the joint space."
  2. In the introduction, the literature review, the author might need to add the comparison (same and different aspects in inverse kinematics and kinematic singularities) of this paper with other papers because the manipulator in this paper is very similar to KUKA LBR 7R (one can treat the parallel three axes as the special case of three intersection axes.) For instance, on page 4, there is no relation between the theta_1 and phi which is similar to the KUKA LBR 7R (the elbow joint).
  3. From page 4 to page 6, the inverse kinematics solution has no closed-form in terms of the input of the end-effector.
  4. The notation s_{2,3,4}, s_{34} and c_{34} etc, are not indicated.
  5. Here is one of the reviewer's main concerns. How to make sure that all the kinematic singularities are found by using the blocks of the Jacobian matrix? Why not take minors of the Jacobian matrix?
  6. The inverse kinematic solution (all possible solutions) at the kinematic singular configuration is not complete. With it, one can know the possibility of avoidance.

Author Response

  1. These phrases were changed according to the reviewer's comment in the abstract.
  2. The difference between the 7-DOF redundant manipulator with three consecutive parallel rotation pairs studied in this paper and the most common SRS configuration 7-DOF manipulator (KUKA LBR 7R) is the rotation pair behind their joint 2. In this paper, the author describes the differences between the two configurations of the manipulator. The kinematic similarity of the two manipulators is that they both have spherical wrist joints without joint distance offset, but their self-motion forms are different. In this paper, the defects of the present research are pointed out for the 7-DOF manipulator with three continuous parallel rotation pairs without distance offset.

  3. In Eq. (3), the homogeneous transformation matrix of the end effector is given first, then the position vector of the end effector is used in Eq. (5), and the orientation matrix of the end effector is used in Eq. (13). Although it is not directly reflected in the results of the inverse kinematics solution, it is an indispensable part of the kinematics solution process.
  4. The subscripts of these symbols represent the sine value or cosine value of the angle sum of the corresponding joint variables. For example, S_{234} represents the sinusoidal value of the sum of angles of joint variables 2, 3 and 4. These symbols have been indicated in the manuscript.
  5. The singular analysis method in this paper is to transform Jacobian matrix into block diagonal matrix by using the idea of elementary transformation, and then to separate singular conditions and concentrate them in a low-dimensional sub-matrix. Then by analyzing singular conditions of the sub-matrix, the singular conditions of the whole Jacobian matrix can be obtained. For this method, all singular conditions can be obtained. The proof of inference is specified in reference [1], and the author adds a reference to this reference. In addition, this method can greatly simplify the calculation, so the author uses the method based on elementary transformation of Jacobi matrix for singular analysis.

  6. The author agrees with the reviewer that the inverse solution in singular parts of kinematics is incomplete or even impossible to solve.

[1] Nokleby S B, Podhorodeski R P. Reciprocity-based resolution of velocity degeneracies (singularities) for redundant manipulators[J]. Mechanism and machine theory, 2001, 36(3): 397-409.

Reviewer 2 Report

Major comments:

  1. In the introduction, the authors thoroughly consider the redundant manipulators and methods of their kinematic analysis, including the singularities. After this discussion, the authors analyze a particular 7-DOF manipulator (with three consecutive parallel axes). However, the authors do not mention why they chose this manipulator and why the entire research is relevant. The authors also do not mention the particular studies on the considered manipulator that may have a gap the authors aim to fill. Thus, the research necessity and its scientific novelty are unclear.
  2. Page 2, lines 55–56. The authors write: “The Jacobian matrix for the redundant manipulator is a non-square matrix, so the singularity conditions can’t be obtained.” One example of the singularity conditions is a rank deficiency of the Jacobian matrix, so the statement above is unclear.
  3. P. 2, Fig. 1. According to the D-H convention [1, p. 27], axis x6 should be orthogonal to z6 and z7, but its direction is different in the figure (see also comment #20). It is also unclear how the authors chose the direction of x7.
  4. P. 3, Table 1:
    • The table has (mm) in its header, while the corresponding parameters are in a symbolic form.
    • There are no explanations for parameters d1, a2, a3, and d4 in the text.
    • Moreover, the following text uses d2 and d3, not a2 and a3.
  5. P. 6, l. 175–176. The authors write the manipulator is in a singular position when the axes of joints 5 and 7 coincide. This should be clarified. Though the values of angles θ5 and θ7 become dependent according to Eq. (15), the rank of the Jacobian matrix (Eq. (18)) still equals six because of manipulator redundancy. Thus, the manipulator, actually, is not in a singular configuration (in its traditional sense).
  6. P. 7, l. 190–192. The authors write: “Since the degree of freedom of the redundant manipulator is greater than 6, the calculation of the Jacobian matrix is very computationally intensive, and it is difficult to obtain singular conditions.” However, there are no explanations. Conversely, we can form this matrix easily, as the authors do in Eqs. (17) and (18). To analyze singular configurations, we can check the matrix rank, as the authors do next. The considered manipulator has seven rotational joints with intersecting or parallel joints, so we can also apply intuitive geometric tools and, for example, check the dimension of a line variety related to the joint axes [2].
  7. P. 8, Eq. (22). Generally, the rank of a block matrix is equal to or less than a sum of the ranks of its blocks [3]. The authors should explain why the equality condition holds here.
  8. P. 10. The authors should explain where the “If…” conditions, which lead to Eqs. (30)–(32), come from.
  9. Logic of section 4.1 is unclear. The authors consider the case when the axes of joints 5 and 7 coincide and calculate the value of angle φ to be avoided. In fact, such a configuration is not a singularity (see comment #5) — axes coincidence is a necessary but not a sufficient condition, as the authors found earlier (Table 2). In this case, i.e., when Eq. (35) is satisfied, we just cannot take an arbitrary value of angle φ. It is determined by Eq. (38), and we can solve the inverse kinematics using relations from section 2.3. Thus, it is unclear why angle φ should not take values according to Eq. (38) as the authors state on p. 12, l. 313–316.
  10. P. 12, Eq. (37). It should be SE instead of SW, according to Fig. 7.
  11. P. 12. The authors should clarify how they obtained Eq. (39) from condition 3.
  12. P. 13, l. 355–356. The authors should explain what the “shortest trip” principle means in more detail (or provide a relevant reference) and why they use this principle.
  13. P. 13, Fig. 9. Manipulator dimensions in the figure are much less than the given ones at the beginning of section 5 (l. 349).
  14. P. 14, Fig. 10. According to Eqs. (40) and (41) and Fig. 9, the end-effector performs a motion with considerable changes of its coordinates (especially in orientation). However, joint angles in Fig. 10 vary only for few degrees.
  15. Sections 5.2–5.3. The authors use maneuverability to measure the manipulator dexterity and estimate its performance. However, Jacobian matrix J the authors use in Eq. (43) is nonhomogeneous in units according to Eq. (18), and the determinant in Eq. (43) can represent the sum of components with various units. It makes the calculation of ω physically meaningless, and the subsequent results become inaccurate. There exist several approaches to handle this problem [4, 5].
  16. P. 16, Eq. (52). The authors apply a polynomial fit to find a polynomial that would approximate their point data. However, the authors do not explain why the obtained polynomial has a third degree. Moreover, according to l. 440, a1 = 0, that makes the polynomial quadratic, but the red line in Fig. 13 has two bending points and cannot be represented by a quadratic curve. Finally, if you evaluate Eq. (52) at t = 100 s (supposing t is in seconds), you will get a value near 8 that differs considerably from Fig. 13.
  17. All the variables used in the paper and their corresponding notations should be explained explicitly in the text. For example:
    • Unit vectors z1, x1, x2, etc., presented in several equations.
    • Rot(x1, π/2) in Eq. (8).
    • AW in Eq. (9).
    • Rotation matrices in Eqs. (13) and (14).
    • Shorthand notations for sine and cosine functions like in Eq. (14). (They differ from Eq. (1).)
    • All notations in Eq. (17).
    • λ in Eq. (19) and the following equations.
    • E, W, and ξ in Eqs. (36)–(38).
    • i in Eq. (42).
    • y and t in Eq. (52).
  18. Some unclear terminology in the text should be clarified. For example, “infinity of inverse kinematics,” “Canadian II” (Canadarm2?), “unfilled rank,” “singular mutation,” “chunking matrix.”

Other comments:

  1. P. 2, l. 66–72. It is unclear, what the authors mean by the singularity avoidance “at the velocity level” and “at the position level.”
  2. P. 2, Fig. 1. It is preferable to picture a manipulator in its general configuration when axis z6 is not parallel with z4.
  3. P. 3, l. 99–100. The authors write: “The rotation of the axes of the last three joints corresponds to a spherical pair.” One should remember that this assumption is correct only when the axes are not coplanar.
  4. P. 4, Fig. 3. There is no designation of point A.
  5. Little curvy arrows near the joints in Figs. 4–7 seem to be unnecessary.
  6. P. 5, Fig. 4:
    • Angle φ should be designated at point S, not W. (Does one need this angle in the figure at all?)
    • According to the text, P’ is an initial plane and P is a rotated one. Given this, angle θ1 is counted clockwise in the figure that differs from the traditional counter-clockwise practice.
  7. P. 5, Eq. (8). Is z2 equal to –n1? (If so, one can avoid unnecessary calculations.)
  8. P. 5, Fig. 5. It should be W instead of W’.
  9. P. 6, Fig. 6. Notations E’ and W’ differ from Fig. 4.
  10. P. 7, l. 192–193. The authors mention the method based on the elementary transformation but do not provide any relevant references.
  11. P. 7. The row order in the Jacobian matrix in Eq. (18) differs from Eq. (17). Moreover, the authors should provide a relevant reference for Eq. (17).
  12. P. 8. The authors permute the rows in the Jacobian matrix, but the new matrix in Eq. (20) has the same notation as in Eq. (19), though the matrices are different.
  13. P. 8, Eqs. (22) and (23). The authors changed notation for the Jacobian matrix from J to S with no explanations.
  14. P. 11, Table 3; P. 13, Fig. 8. Similarly to comment #20, it is preferable to picture the manipulator when angle θ5 is unequal to zero.
  15. P. 11, l. 289. It should be “Table 4” instead of “Table 3” in the table header.
  16. P. 12, Fig. 7. The figure uses notations zS and xS instead of x1 and z1 used earlier with no explanations.
  17. P. 13, Table 5. In the figures, angles θ2 and θ5 both equal zero. Again, it is preferable to show the manipulator when these angles are distinct from zero, and only singular conditions are satisfied.
  18. P. 14, l. 371. The authors should provide a relevant reference for the term “maneuverability” and Eq. (43).
  19. P. 14, Eq. (44). Operator “det(ω)” looks strange since ω is a scalar.
  20. P. 14–15, Eq. (45) and (47). It is incorrect to equate a vector to a matrix.
  21. P. 14–16. There are no units in Eqs. (46) and (48) and in y-axis in Figs. 11, 12, and 14 (see also comment #15).
  22. P. 15, l. 393. The authors use the term “horizontal singularity” previously unmentioned.
  23. P. 15, l. 401–402. The authors write: “It can be seen from Eq. (47) that the distance between S and W of the manipulator satisfies the singularity condition of position.” Actually, it is difficult to see from Eq. (47) that this condition is satisfied.
  24. P. 16, l. 427. The authors use the term “initial criterion of singularity” with no explanations.
  25. P. 16, Fig. 13. There is no information about the color bar (its corresponding variable with units) in the figure.
  26. P. 16, Figs. 13 and 14. The authors use the terms “evaluation index” and “operability index” in the figures captions with no explanations (do they mean “maneuverability” used earlier?).
  27. P. 16, l. 440. Polynomial coefficients have no units.

With the comments above, the paper needs significant improvements, and I cannot recommend the presented work for publishing in the Journal.

References:

  1. Waldron K.J., Schmiedeler J. (2016) Kinematics. In: Siciliano B., Khatib O. (eds) Springer Handbook of Robotics. Springer, Cham, pp. 11–36.
  2. Dandurand A. (1984) The rigidity of compound spatial grids. Structural Topology. Vol. 10, pp. 41–56.
  3. Marsaglia G., Styan G.P.H. (1974) Equalities and inequalities for ranks of matrices. Linear and Multilinear Algebra. Vol. 2, iss. 3, pp. 269–292.
  4. Patel S., Sobh T. (2015) Manipulator performance measures – a comprehensive literature survey. Journal of Intelligent & Robotic Systems. Vol. 77, pp. 547–570.
  5. Rosyid A., El-Khasawneh B., Alazzam A. (2020) Review article: performance measures of parallel kinematics manipulators. Mechanical Sciences. Vol. 11, pp. 49–73.

Author Response

  1. The 7-DOF redundant arm with three consecutive parallel rotating pairs and the most common SRS configuration 7-DOF manipulator studied by the author. The difference between them is that the rotating pairs behind their joint 2 are different. At present, there are more researches on SRS configuration manipulator and more practical industrial applications. And the mechanical arm with three continuous parallel rotating pairs is also a kind of optimum structure mechanical arm. The author describes the difference between the two configuration mechanical arms. At present, the mechanical arm with three continuous parallel rotating pairs has three continuous parallel rotating pairs. The research on the mechanical arm mostly focuses on the space manipulator, but the research of mechanical arm with spherical wrist joint without joint distance offset is very few. Therefore, the author has carried out relevant research on it, and the mechanical arm with this configuration can be further developed. In this paper, the defects of the present research are pointed out for the 7-DOF manipulator with three continuous parallel rotation pairs without distance offset.
  2. What the author wants to express here is that when the Jacobian matrix is a non-square matrix, the singular conditions cannot be obtained by simply finding the matrix determinant. The author's statement here is not clear enough, and now it has been described more clearly.
  3. According to the D-H convention, axis x6 should be perpendicular to Z6 and z7, which has been corrected in the figure 1. For the convenience of calculation, let axis x7 and axis x6 have the same direction.
  4. According to the reviewer's comments, The units of symbolic parameters in Table 1 have been deleted. the parameters d1, d2, d3 and d4 are d1 = |OS|, d2 = |SA|, d3 = |AE|, d4 = |EW|, and they are marked in figure 1. The parameters a2 and a3 in the text should be d2 and d3, which have been corrected in the text.

  5. The axes of joint 5 and joint 7 of the manipulator coincide, and it can not be concluded that the manipulator is in a singular position, which has been deleted by the author in this paper. In fact, this condition alone is not sufficient to determine whether the manipulator is in a singular configuration.
  6. The author's statement here is incorrect and has been corrected. The meaning the author wants to express is actually because the Jacobian matrix is not a square matrix and it is difficult to obtain the singular conditions by means of whether the matrix determinant is zero. Therefore, the author uses the singular analysis method based on the elementary transformation of the matrix to obtain the singular conditions. The method of using intuitive geometric tools to analyze singularities mentioned by the reviewer is also possible, but it may be difficult to intuitively imagine some singularities using this method.
  7. According to the theorem (8.11) of block matrix rank in the literature [3], and S12 is full rank matrix, so the equality condition of Eq. (22) can be obtained, and a relevant reference has been added in this paper.
  8. Eqs. (30) and (32) come from Eq (26), which is one of the conditions for determinant equal to 0 and one of the conditions for rank deficiency of Jacobian matrix, that is, the condition for rank deficiency of Jacobian matrix. Through this condition, the situation that S6 is not equal to 0 is analyzed.

  9. According to the reviewer's comments, due to logic problems in Section 4.1, the author has made a lot of changes to it. The coincidence of the 5-axis and 7-axis of a joint (i.e. s6=0) is one of the conditions under which the singularity can be determined. It is not possible to determine whether the singularity is singular unless the condition satisfies s3=0 or s5=0 at the same time. In the original manuscript, the author did not consider the influence of the position and orientation of the end-effector on the singularity, i.e. through Eqs (34) and (35). In the revised version, it is assumed that when s3=0, the redundant angle corresponding to the singular configuration can be deduced according to the position and orientation of the given W point, and the EW vector can be obtained to judge the relationship between the axial vector of joint 7 and the EW vector. If the two axes are in line, it can be said that the robot arm is in the configuration corresponding to singular condition 1. In motion planning, singularity can be avoided as long as the redundant angle of singular configuration is not taken. However, for singular condition 2, although singularity can also be avoided by changing the redundant angle φ, it is impossible to judge whether the manipulator is in singular configuration under singular condition 2 only by knowing the position and orientation of W point, and additional conditions are required to judge.

    For singular condition 3, in order to better understand the configuration characteristics of the mechanical arm under this condition, the author supplements a diagram of geometric illustration of singular condition 3. In this paper, for singular condition 4, these four configurations have the same geometric characteristics, as shown in Table 5. By judging the relationship between SW and link length d2, d3 and d4, it can be determined whether the manipulator is in this kind of singular configuration. As is done by singular condition 1 and singular condition 2, this kind of singularity can be avoided as long as the corresponding redundant angle of singular configuration is not taken in motion planning.

  10. According to the reviewer's comments, SW is now changed to SE in Eq. (37).
  11. According to the reviewer's comments, the geometric meaning of singular condition 3 is that the projection of point W in the horizontal direction x is 0, that is, the end of the manipulator is on the axis of joint 1.
  12. Due to the multi solution problem of the manipulator, the "shortest travel" is reasonably selected to ensure the continuity of joint motion, that is, the possible solution with the smallest joint angle difference from the previous time is selected, so as to ensure a smooth transition of joint motion between the two times.

  13. The author corrects the size of the manipulator at the beginning of section 5.

  14. In Figure 10, the unit of joint angle θ is radian, which has been corrected by the author.
  15. The manipulability index is scale or units dependent. The value of the Jacobian determinant depends on the choice of the physical units used; therefore, the manipulability index will have different values for different units used to represent the link lengths and joint angles. This may make the calculation of ω meaningless in physics, especially for the dexterity measurement of the manipulator in the workspace. However, in this manuscript, the author only uses operability to identify the singularity rather than measure the dexterity of the manipulator. According to the definition of operability, when it is located at the singular point, ω = 0, and the corresponding manipulator is in a singular configuration, The following types â…  and â…¡ can also prove this.

  16. The author's polynomial fitting expression is not derived correctly. After careful consideration, the expression is meaningless here, so the author decides to delete it without affecting the logical relationship of the context.

  17. According to the reviewer's comments, all variables and their corresponding symbols used by the author in this paper are clearly explained.

  18. These terminologies were changed according to the reviewer's comment. The term of "Infinity of inverse kinematics" means that the number of inverse kinematics solutions is infinite. The term of “Canadian II” was changed to Canadarm2. The term of "Unfilled rank" refers to the rank deficiency of Jacobian matrix, which has been corrected now. The term “singular mutation” indicates that there is a peak corresponding to the value λ at the singularity, which has been modified to make it easier to understand. The term "chunking matrix." is rarely used. Now it has been changed to block matrix.

  19. The authors have rewritten and explained the two terms, which are now changed to velocity-based singularity avoidance methods and position-based singularity avoidance methods. The velocity-based singularity avoidance method generally determines whether singularity occurs by calculating the rank of the Jacobi matrix as a way to change the motion of the manipulator, while the position-based singularity avoidance method obtains the singularity condition of the mechanism by the singularity analysis of the mechanism, and what type of singularity occurs at a certain position of the robot arm is known.

  20. The author thinks that if the Z6 axis is not parallel to the Z4 axis, the coordinate system will look more chaotic, and the current manipulator configuration will show the manipulator coordinate system more clearly.

  21. According to the reviewer's comments, the premise of "only when the axes are not coplanar" was added in the article.

  22. According to the reviewer's comments, point A is now designated in the figure 3.
  23. According to the reviewer's comments, little curvy arrows near the joints in Figs. 4-7 have been deleted.

  24. According to the reviewer's comments, angle φ should be designated at point S in Fig. 4 have been deleted. According to the traditional counter-clockwise practice, the initial plane P 'and rotation plane P are adjusted in the Fig 4.

  25. The z2 vector in this paper is equal to -n1 and the author has corrected it in this paper.
  26. The point W' is now changed to W in the Fig 5.
  27. The symbols E 'and W' in Fig. 6 have different meanings from Fig. 4. Points E and E' in Fig. 6 represent two possible cases of point E, i.e. point E is above and point e 'is below, respectively, while points E and E' in Fig. 4 represent the case of the same point E in two different planes. The author has further explained the meaning of point E and point E' in Figure 6.

  28. A reference[1]  has been provided to the method based on elementary transformation.
  29. The row order of the Jacobian matrix in Eq. (17) is now corrected. A reference [2] has been added.
  30. The new matrix in Eq. (20) is distinguished from the matrix notation in Eq. (19).
  31. The notation S of the Jacobian matrix in Eqs (22) and (23) should be J after elementary transformation, which has been modified in this paper.

  32. According to the suggestion of the reviewer, the mechanical arm whenθ5 is not equal to 0 is drawn in the figure and table.
  33. The table header of “Singularities corresponding to condition 2” was changed to Table4.
  34. As the author has made a lot of changes in sections 4.1 and 4.2, the original Figure 7 has been deleted in the revised draft.
  35.  According to the suggestion of the reviewer, the figure of the manipulator when θ2 and θ5 are not equal to 0 and only satisfy the singular condition is drawn in Table 5.
  36. A reference [3] has been provided for the term "maneuverability" and Eq. (43).

  37. Operator “det(ω)” was correctly changed to “det(J·J^T)”.
  38. According to the reviewers' opinions, the author distinguishes the vectors and matrices in the equation with different symbols.
  39. The units in Eqs. (46) and (48) are radians. The variables in Figure 11, 12, and 14 are related to the manipulability index ω, which is dimensionless and therefore has no units.
  40. The author mistakenly uses this term, which has been corrected and restated in the text. Its correct meaning is that whether singularity will occur can be judged according to the end-effector position and orientation relationship by using the above singularity judgment method for type 1.
  41. Eq. (47) in the original draft is now Eq. (52) in the revised manuscript. According to Eq. (44), it can be concluded that this condition is met, and the author explains it in the later paragraph of Eq (52).
  42. The term used here is incorrect and has been corrected. The notation xe1 and xe2 are the pose vectors of the end effector in case 2, which represent two types of singularity respectively.
  43. The color bar in the figure represents the value of the maneuverability index ω, which is a dimensionless value. The meaning of its color is explained in the later paragraph of Figure 13.
  44. These terms have the same meaning as " maneuverability ". The author has corrected it in Figs. 13 and 14.
  45. The polynomial has been deleted by the author (see also comment #16).

[1] 2013 IEEE/RSJ International Conference on Intelligent Robots and Systems (IROS)November 3-7, 2013. Tokyo, Japan

[2] The Mathematics of Coordinated Control of Prosthetic Arms and Manipulators

[3] Yoshikawa T. Manipulability of robotic mechanisms. The international journal of Robotics Research, 1985, 4(2): 3-9.

Round 2

Reviewer 2 Report

Major comments:

  1. Page 2, lines 76–83. The authors have added material about two 7-DOF manipulators (SRS and the one with a distance offset) and several related references. However, the authors do not provide any information on their considered manipulator, except for “There is a little research on kinematics and singularity…”. But what about any particular studies? Or do the authors propose a novel manipulator, unstudied earlier? Such information is crucial and should be presented explicitly in the text. It would help to clarify the research necessity and its scientific novelty, which are still poorly stated.
  2. P. 7, l. 220–222. In the response, the authors say they have corrected their statements. However, the material is the same as in the previous version of the manuscript. Please check.
  3. P. 11. In the response, the authors say that the “If…” conditions come from Eq. (26). But the matrices in Eqs. (26) and (29) are different and relate to different cases for s6. In the text, it is still unclear (at least for the reviewer) why we should refer to Eq. (26).
  4. P. 12–13. As the reviewer understands, the authors first find Eq. (35) corresponding to condition s6 = 0. Next, the authors calculate the redundancy angle that relates to condition 1 (Table 2) and should be excluded for singularity avoidance. Now, it seems clear and reasonable. However, the purpose of the material in p. 13 is unclear (Eq. (38), (40), and (41)). Isn’t it enough to check Eq. (35) and avoid angle φ from Eqs. (37) or (39) if Eq. (35) is satisfied?
  5. P. 16, l. 423–427. The authors first write that “the amount of motion of each joint is minimized,” but later consider the minimization of only the second joint.
  6. Section 5.3. The reviewer agrees with the authors that the manipulability index can be used to check the singularities without modifying the Jacobian matrix. However, when the authors calculate this index along the trajectory (Fig. 14), the nonhomogeneous units can affect the result. Thus, it is still inaccurate to state that “the manipulator have the best configuration and flexibility” (p. 19, l. 524–525).

Other comments:

  1. P. 1, l. 37. It should be “…number of inverse kinematic solutions…”.
  2. P. 1, l. 40. Shouldn’t it be “kinematic control” instead of “kinetic control”?
  3. P. 2, l. 55. The authors use the acronym “SRS” without explanations: it appears later on l. 76.
  4. P. 2, l. 71–75. As the reviewer understands, analyzing singularities for a redundant manipulator at the position level (used in the manuscript) relates to analyzing the number of inverse kinematic solutions. For example, this number becomes finite, and we can explicitly find the joint coordinates to be avoided. Is this what the authors mean? If so, it is still unclear in the text.
  5. P. 2, l. 80–81. Terms “joint 2 rear rotating pair” and “distance offset” are unclear in the context.
  6. P. 3, Fig. 1. The authors should check the direction of axis x5 according to D-H rules. Shouldn’t it have an opposite direction?
  7. P. 5, l. 167–168. Formulae (8) do not use notations z1 and Rot mentioned in the text.
  8. P. 6, Fig. 6. Does the figure need notation W’ if it coincides with W?
  9. P. 7, l. 217–218. The authors use the term “generalized velocity” with no explanations.
  10. P. 7, l. 227; p. 22, l. 632–633. Reference [36] doubles [23].
  11. P. 8, Eq. (17). It is incorrect to multiply a vector by a vector on the right side of the equation. There should be a matrix with columns corresponding to the expression in the brackets.
  12. P. 8, l. 236–237. ω and υ should be typed in bold as in Eq. (17); notation θi is not used in Eq. (17) and should be changed.
  13. P. 11, l. 297 and Eq. (30). It is inaccurate to transform matrix 5S21 from 3×4 to 3×3 keeping the same notation.
  14. P. 13. According to Fig. 7, it should be EW instead of EW’ in the text and formulae on the page.
  15. P. 13, Eqs. (36), (37), and (39). SW should be typed in bold.
  16. P. 13, Eqs. (38) and (40). Does SE in the equations mean |d2 ± d3|?
  17. P. 13, l. 368. The authors use the term “approach vector” with no explanations.
  18. P. 14, Fig. 8. Actually, the figure does not relate to singular condition 3 (as mentioned in its capture) but just demonstrates a fragment of the manipulator.
  19. P. 16, Eq. (47). Index i has no appropriate description.
  20. P. 17, l. 456. The authors use the acronym “RPY” without explanations.
  21. P. 17, l. 462. The authors use the previously unmentioned term “singularity index.” Is it the name of λ?

With the comments above, the paper still needs improvements.

Author Response

  1. According to the reviewer's comments, the author's statement here is not clear enough. The 7-DOF redundant manipulator with three continuous parallel axes studied by the author is also one of the better configuration of 7-DOF manipulator. Another optimum configuration is the SRS configuration 7-DOF robot arm. There have been many related studies on the kinematics and singularity of this configuration, but there are few related studies on the robot arm studied by the author. Therefore, the author uses the intuitive geometric method to solve the inverse kinematics and propose a position-based singularity analysis method for singularity analysis.

  2. The author forgot to revise it in the previous revision, which has now been corrected. (P. 7, l. 222–224.)

  3. Consider Eq. (26), the first column elements of matrix A11 and A12 in Eq. (29), it can be seen that they both have a common factor term d2c2+d3c23+d4s234. The discussion on whether the factor term is zero for classification can facilitate the calculation of singular conditions, when the factor is equal to zero, the first column of matrix of Eq. (29) all become zero, at this time the matrix becomes a square matrix, and the singular conditions can be obtained conveniently with the determinant, but at this time The determinant of the square matrix is zero, therefore, s6≠0 and d2c2+d3c23+d4s234=0 is the singular condition of the matrix, according to Eq. (26), we can know that s6=0 and d2c2+d3c23+d4s234=0 is also the singular condition of the matrix, so we can say that as long as d2c2+d3c23+d4s234=0 is the singular condition of the matrix. In another case, when the factor term is not equal to zero, the matrix singularity condition when d2c2+d3c23+d4s234≠0 can be obtained by elementary transformation of the matrix of Eq. (29). (P. 11.I.302-308.)

  4. Satisfying Eq. (35) can only show that the axis of joint 7 lies in the main plane, i.e., s6 is equal to zero. According to singularity condition 1, it is known that the singularity is related to s6 and s3, as shown in Figure 7. However, for singularity condition 2, which is related to s6 and s5, such singularities cannot be determined if only the end position and attitude are known, and further judgment is required to do so, which has been explained by the authors in the paper.

  5. In the previous version, the author's statement here was not clear enough, but now it is further explained in the paper. From the perspective of energy saving, it is assumed that joint 2 should be kept stationary as far as possible during the movement, and the inverse kinematics of other joints should be selected according to the principle of "the shortest trip". (P. 15.I.424-428.)

  6. The author's statement here is too absolute. Now a new statement is changed. The manipulability index ω will make the manipulator have better configuration and flexibility. (P. 19.I.532-534.)

  7. According to the comments of reviewers, the author made corrections in the revised manuscript. (P. 1.I.37.)

  8. According to the comments of reviewers, the author made corrections in the revised manuscript. (P. 1.I.40.)

  9. The acronym SRS means that the manipulator is equivalent to a spherical joint, revolute joint, and spherical joint. The author explains it where it first appears in the text. (P. 2.I.55.)

  10. The position-based singular avoidance method means that the position of the end-effector and the position of the joint angle are used to determine whether the manipulator is in a singular configuration. (P. 2.I.76-78.)

  11. The term of “joint 2 rear rotating pair” means that the axis direction of joint 3 is different, which has been corrected in this revising paper. The term of “distance offset” means that the axes of the spherical joints do not intersect, such as the axes of last three joint near the end-effector of UR manipulator. (P. 2.I.84-85.)

  12. According to D-H rules, when the xi of x-axis of the coordinate system {i} coincides with ai, both positive and negative axis directions of xi can be selected.

  13. According to the comments of reviewers, the terms not mentioned in the text have been deleted. (P. 6.I.171-172.)

  14. The notation W’ has been deleted in the Figure 6.

  15. The term of “generalized velocity” means that the linear velocity and angular velocity. (P. 7.I.219-220.)

  16. The reference [36] has been deleted.

  17. Eq. (17) has been corrected, and Eq. (18) has been supplemented, which represents the calculation formula of column ith of Jacobian matrix. (P. 8.I.232-242.)

  18. ω and υ have been changed to bold, notation θi has been changed to the notation of joint velocity vector.

  19. Eq. (29), Eq. (30) and Eq. (31) are different and are distinguished by the upper right corner mark of the matrix symbol. (P. 11.I.297-304.)

  20. According to the reviewer's comments, EW' in the text and formula has been changed to E'W. (P. 12-13.I.357-366.)

  21. The Eqs. (37), (38), and (40) in the revised version, SW is changed to bold.

  22. When in singular condition 1, SA is collinear with AE, so SE means | D2 ± d3|. Figure 7 (b) was wrong in the previous version, and now it is corrected, which means that SE is equal to |d2 - d3|.

  23. The term “approach vector” has been changed to the axis vector of joint 7. (P. 13.I.370)

  24. As far as the author is concerned, Eq. (43) means that the projection value of point W in the horizontal direction x. When the Eq. (43) is equal to zero, it is actually equivalent to singular condition 3.

  25. The symbol i represents the ith waypoint among many discrete points of a trajectory.

  26. The RPY angle means that roll angle, pitch angle and yaw angle. (P. 17.I.463)

  27. As the reviewer understands, λ is the name of the singularity index, which is explained below Eq. (50). (P. 16.I.457-458)

Round 3

Reviewer 2 Report

The authors have successfully reviewed all the comments.